# Fatigue Detection during Sit-To-Stand Test Based on Surface Electromyography and Acceleration: A Case Study

**DOI:** 10.3390/s19194202

**Published:** 2019-09-27

**Authors:** Cristina Roldán Jiménez, Paul Bennett, Andrés Ortiz García, Antonio I. Cuesta Vargas

**Affiliations:** 1Instituto de Biomedicina de Málaga (IBIMA), Grupo de Clinimetría (F-14); 29010 Málaga, Spain; cristina.roldan005@gmail.com; 2School of Clinical Science, Faculty of Health Science, Queensland University Technology, Queensland, Kelvin Grove QLD 4059, Australia; p.bennett@qut.edu.au; 3Andrés Ortiz, Communications Engineering Department. ETS Ingeniería de Telecomunicación. University of Málaga, 29071 Málaga, Spain; aortiz@ic.uma.es; 4Department of Physiotherapy. University of Malaga, Faculty of Health Sciences, 29071 Malaga, Spain

**Keywords:** electromyography, kinematics, fatigue, acceleration, sit-to-stand, motion analysis

## Abstract

The latest studies of the 30-second sit-to-stand (30-STS) test aim to describe it by employing kinematic variables, muscular activity, or fatigue through electromyography (EMG) instead of a number of repetitions. The aim of the present study was to develop a detection system based on acceleration measured using a smartphone to analyze fatigue during the 30-STS test with surface electromyography as the criterion. This case study was carried out on one woman, who performed eight trials. EMG data from the lower limbs and trunk muscles, as well as trunk acceleration were recorded. Both signals from eight trials were preprocessed, being averaged and temporarily aligned. The EMG signal was processed, calculating the spectral centroid (SC) by Discrete Fourier Transform, while the acceleration signal was processed by Discrete Wavelet Transform to calculate its energy percentage. Regarding EMG, fatigue in the vastus medialis of the quadriceps appeared as a decrease in SC, with a descending slope of 12% at second 12, indicating fatigue. However, acceleration analysis showed an increase in the percentage of relative energy, acting like fatigue firing at second 19. This assessed fatigue according to two variables of a different nature. The results will help clinicians to obtain information about fatigue using an accessible and inexpensive device, i.e., as a smartphone.

## 1. Introduction

The sit-to-stand test (STS) represents a common movement performed in activities of daily living. This test makes it possible to quantify the functional activity of getting up from a chair [1], and assesses lower body strength [2]. STS implies skills such as coordination, control of balance, and stability [3], and it is also biomechanically demanding. Therefore, STS is essential in the assessment of patient independence [4,5], and studies using this test have been performed over several decades [6,7]. One of its variations is the 30-second STS (30-STS), in which the subject has to alternate sitting and rising from a chair for 30 seconds as quickly as possible. The number of repetitions completed provides quantitative information with which to evaluate functional fitness levels [8], and has been used in the rehabilitation field [9]. Recently, several studies have highlighted the importance of kinematic parameters to provide qualitative information about how the motion is carried out [10]. In this field, inertial units and accelerometers have been validated in STS variants in order to measure movement duration [11,12]. More recently, studies have focused on analyzing kinematic variables provided by these devices, which have allowed researchers to differentiate between healthy adults and to identify different levels of frailty [13], fallers and non-fallers [14], and young and older adults [15]. Kinematic variables such as velocity or acceleration in 30-STS are more sensitive in discriminating between some populations than the number of cycles completed [13]. Also, the number of repetitions can be complemented by kinematic values. Providing information about how the movements were carried out yields quality information, which improves rehabilitation [16]. More specifically, acceleration provides an accurate and simple method to describe kinematics, assessing spatiotemporal characteristics during STS [12].

Besides kinematics, electromyography (EMG) also provides additional information from the STS. In this field, several variables have been analyzed, such as muscle activation in asymptomatic subjects, in order to identify how muscles weakness can limit the test [17], or to measure muscular demands, both with and without the use of kinematic devices [18]. Furthermore, neuromuscular activity has been studied in different physical environments [19].

One of the most studied variables by EMG is fatigue, as EMG is considered the gold-standard for measuring fatigue [20,21,22,23]. However, changes in the EMG signal represent modifications in muscular membrane properties and metabolic conditions [24], while fatigue is a multidimensional concept that involves psychological and physiological dimensions [25]. On the one hand, the psychological or subjective dimension of fatigue refers to perceptions of fatigue, and is composed of psychological factors such as perceptions of effort, expectations, motivation, arousal, or mood [26]. This subjective dimension is usually measured by patient-reported outcomes [27]. On the other hand, the physiological dimension of fatigue can be observed in both the central and peripheral system domains [25]. In the peripheral system, muscle, nerves, and glycogen stores comprise physiologic factors [26]. Muscle fatigue study can be conducted by measuring EMG signals using surface electrodes to search for specific patterns. However, information related to fatigue is not present in the signal amplitude, but in the frequency spectrum. In fact, the most popular methods for fatigue detection using EMG signals are based on the measurement of the mean frequency and median frequency in the signal spectrum [28,29,30,31]. Specifically, [28,29,30] show that muscle fatigue results in a downward shift of the frequency content of the EMG signals. As a result, fatigue can be detected by observing a downward shift in the mean frequency (MNF) or median frequency (MDF) values. 

Related to the present work, fatigue during the 30-STS test has been previously analyzed in the lower limbs and trunk muscles by EMG [32]. Muscle fatigue is defined as a state of depressed responsiveness resulting from activity [33]. Hence, it constitutes a very important parameter, providing motion quality information because it affects kinematic parameters in the lower limbs [34,35,36,37,38]. Regarding kinematics, acceleration during STS has been related to variables such as peak power [39]. Previous studies have also highlighted the impairment of movement due to fatigue. This even includes what is known as “biomechanical fatigue”, referring to fatigue that is measured by speed and acceleration, and is closely related to kinematics [22]. In this field, variables such as velocity [40] or acceleration [41] can differentiate between fatigued and non-fatigued training exercises. However, these kinematic differences were analyzed by comparing execution before and after a fatiguing activity, and the variation in joint angles was used as an indicator of fatigue rather than the execution itself. Only one previous study considered a reduction in the percentage of trunk velocity with respect to the maximum value recorded during the first repetitions of STS. This study assessed fatigability in older women using a linear encoder fixed with a belt on the hip which measured the distance per time unit [42].

Although muscular fatigue during 30-STS has been studied previously using surface electromyography [32], and previous studies have focused on kinematics during this test [42], no study has analyzed fatigue through 30-STS acceleration measurements. Hence, the aim of the present study was to study fatigue using a detection system based on acceleration measured using the sensors embedded in a smartphone, as well as surface electromyography, in a subject undertaking the 30-STS test. 

## 2. Materials and Methods

### 2.1. Subject

A single subject performed all trials to avoid intersubject variability. The subject was a 25-year-old woman, 1.60 m in height, 83 kg, and with a BMI of 32.42 Kg/m^2^. The chosen woman was obese and sedentary to ensure fatigue during the test. The subject did not suffer from any cognitive disorders, musculoskeletal, bone or joint injury, and was not taking any medication. The subject was recruited from the Faculty of Health Sciences (University of Málaga, Spain). 

Ethical approval for the study was granted by the Ethics Committee of the Faculty of Health Sciences, University of Malaga. The study complied with the principles laid out in the Declaration of Helsinki. The subject provided written consent to participate after she had been given information about the study, that participation was voluntary and she could withdraw at any point.

### 2.2. Apparatus

Operative variables were obtained through an electromyograph (Megawin 3.0.1., Mega Electronics Ltd, Kuopio, Finland) and a Smartphone (Samsung Galaxy J5, Samsung Electronics Co., Ltd. Suwon, Korea) providing electromyograhic and kinematic measurements, respectively. Regarding electromyography, the apparatus consisted of 8 independent channels which made it possible to measure 8 different muscles (see Figure 1). Each EMG channel had three outputs, which allowed us to use 3 electrodes per muscle. Three square surface electrodes (Lessa, Barcelona, Spain, 2.5 cm, Ag/AgCl) were placed as described by Rainoldi [43]. To ensure good adhesion, the skin was washed with alcohol and shaved very gently, avoiding increased blood flow to the area. The muscle activity of the gastrocnemius medialis (GM), the biceps femoris (BF), the vastus medialis of the quadriceps (VM), the abdominal rectus (AR), the erector spinae (ES), the rectus femoris (RF), the soleus (SO), and the tibialis anterior (TA) on the subject’s dominant side were recorded at a frequency of 1000 Hz. The Maximal Voluntary Contraction test (MVC) was performed for each muscle, as proposed by Perotto [44]. Functional muscle testing for the MVC test was carried out using the method described by Daniels [45]. The subject was previously trained to obtain MVC by biofeedback observations of her muscle activity on the computer screen. Hence, the electromyographic descriptive variables were as follows: real activation (µV), calculated from the maximum value minus the minimum value of MVC test to normalize the signals; average muscle activation (µV); MVC percentage (%), normalized to represent the percentage of activation of each muscle relative to its real activation value; and muscle involvement in motion (%), representing the activation percentage of each muscle with respect to the total muscular activity signal recorded during the STS motion. Regarding kinematic, linear acceleration, expressed in m/s^2^, was measured along three orthogonal axes (*x, y, z*) using the Samsung Galaxy J5 accelerometer, which was snugly secured to the subject by a neoprene fixation belt over the sternum (see Figure 1). 

The reliability and validity of the accelerometer embedded in a smartphone placed in sternum have been previously tested [46]. The application used to obtain kinematic data was Sensor Kinetics Pro (Innovetions Inc, United States), available at the Google Store. The data for each trial was transmitted as an email for analysis and postprocessing. The data-sampling rate was set at 25 Hz. Linear acceleration was calculated in each axis of space by resting the minimum peak to the maximum peak. The X axis lay in the lateral direction, Y was in the vertical pointing upwards, and the Z axis was in the front-back direction, representing lateral, up-and-down, and anterior-posterior acceleration, respectively. Acceleration was also calculated with regard to the norm of the resultant vector (Nrv) of the three axes of movement: $$\mathrm{Nrv}\;=\sqrt{x^{2}+y^{2}+z^{2}}$$

The values for muscle activity were recorded simultaneously using laptop hardware (Megawin 3.0.1., Mega Electronics Ltd, United Kingdom). A DV Trigger [Mega Electronics Ltd] device connected to the laptop temporarily synchronized both measures by placing a marker at the start of smartphone recording, as well as at the beginning and at the end of the test. 

### 2.3. Experimental Setup

The investigator explained the task in a clear and concise manner, and the start and end of the task were marked by an audible signal. Tasks were performed in the Laboratory of Human Movement at the Faculty of Health Sciences (University of Málaga). Each trial was carried out by the same subject on eight different days with a period of 1 week between each measure and at the same time of day. Also, the subject was instructed not to eat for 2 hours prior to the test to avoid an influence on muscle reserves, such us creatine phosphate, employed in short, high-intensity tasks [47].

The 30-STS test was performed beginning in the standing position, with the feet the same distance apart as the hips, and the upper limbs crossed over the anterior of the body, with bent elbows to avoid impulses (see Supplementary Materials). Given a signal from the investigator, the subject had to sit and rise from a 43-cm-high chair as quickly as possible through the whole range of motion for 30 S.

### 2.4. Preprocessing

EMG and inertial data were preprocessed in the following way: Raw data from both signals were rectified and then filtered using a low-pass 4th order Butterworth filter with a cut-off frequency of 300 Hz. Through the rectification process, the average frequency was subtracted so that the mean value of the signal was zero. In the filtering process, the low-pass 4th order Butterworth filter removed interference or components which were outside the bands of interest.

Once the EMG and acceleration signals were rectified and filtered, the signals obtained during eight repetitions of the experiment were averaged, and this averaged signal was used for all further analysis. This requires the temporal alignment of signals acquired during the 8 different experiments, which was addressed by taking the maximum value of cross-correlation with respect to the first acquisition as a reference. This way, further processing was carried out using the averaged signals.

### 2.5. Data Processing

Classical EMG processing used to identify muscle fatigue using surface electrodes is based on the detection of a decrease in the MDF of the EMG spectrum [48,49,50,51] during an isometric muscle action. In fact, the MF and spectral centroid (SC) are both considered measures of central tendency of the spectral distribution [52]. In order to identify muscle fatigue in EMG in a dynamic condition, the SC (which is related to the “center of mass” of the discrete Fourier transform (DFT) spectrum) was computed in each overlapped (0.5 overlapping factor) Hanning window.

### 2.6. Data Analysis

Descriptive statistics, measures of central tendency, and dispersion of the study variables were calculated. The mean and SD of the eight different measures were calculated for real activation, average activation, MVC percentage, and muscle involvement, as well as for acceleration in both axes and Nrv. 

The SC is defined as:$$SC^{n}=\frac{\sum_{k=1}^{N}k\;f^{n}\left[k\right]}{\sum_{k=1}^{N}f^{n}\left[k\right]}$$
where *f^n^*[*k*] is the amplitude corresponding to the *k*^th^ bin in the DFT spectrum of the *n*^th^ window [48]. 

Moreover, a further step to clearly identify changes in the spectral centroid was taken by computing the difference in SC between consecutive windows. In order to determine the intrasubject variability, both standard error of the measurement (SEM) and standard deviation (SD) from the slope of the spectral centroid were calculated. The SEM was calculated using the formula SEM = s$\sqrt{1-r}$, where s is the mean and SD of eight measurements and r is the reliability coefficient for the test and Pearson’s correlation coefficient between values. 

Inertial data processing is performed in the frequency domain, and consists of computing the average power on different subbands by means of the Discrete Wavelet Transform, computed in overlapped windows (overlapping factor = 0.5). Specifically, we computed the variance of the approximated and detailed coefficients from five levels using the Daubechies *db4* as the mother wavelet. In addition, the calculation of the energy ratio of the approximated coefficients A_i_ and detailed coefficients D_i_ helps to find a more clear pattern [53]. At level *i,* these are computed as *EDR_Ai_ = A_i_A_i_^T^/E_T_* and *EDR_Dj_ = D_i_D_j_^T^/E_T_* (*j* = 1, ..., *i*) for the approximation and detailed coefficients, respectively [49], where
$$E_{T}=A_{i}A_{i}^{T}+\sum_{j=1}^{i}D_{i}D_{j}^{T}$$

Fatigue detection from inertial data has been addressed by analyzing the relative energy for each window, as computed from the DWT coefficients at different levels. This way, a breakdown point in the average power can be computed from Level 4 and 5 coefficients. 

## 3. Results

The muscle with the highest real activation value during the MVC test was TA, followed by VM. Muscles with the highest average level of activation during STS were TA followed by RF, SO, and VM. When signals were normalized (MVC percentage), the highest values were found in RA, RF, SO, and GM. Regarding the distribution of muscle involvement in STS, the highest percentages were found in the TA, RF, SO, and VM muscles (see more details in Table 1).

SC variation from the EMG signal is represented in Figure 2. Each channel represents the SC from averaged 8 measurements after being rectified, filtered, and temporarily aligned. Regarding intrasubject variability, the SEM and SD values from the 8 averaged measures from VM were 0.05 and 0.15, respectively. Changes in slope were observed in all channels during the test. However, in channel 3, which corresponded to the VM muscle, a highlighted frequency shift between seconds 10 and 15 could be observed. The area under the curve for the VM muscle signal decreased at second 12, with a slope of 12%. 

Fatigue is usually detected by a decrease in the MDF of the EMG spectrum. In the case of VM muscle (channel 3), the MDF was 51.5Hz at the beginning and 46.5 Hz at the end of the 30-STS. SC can be also used to detect fatigue, since a decrease in the MDF implies a decrease in the SC. To show this, Figure 3 depicts the variation of both the SC and MDF of the EMG signal of the VM muscle (channel 3) measured during 30-STS.

The highlighted frequency shift found in SC in VM muscle at second 12 (Figure 2, channel 3) temporarily corresponds to a descriptive acceleration value of 0.98 m/s^2^. More details are shown in Figure 4.

As far as acceleration was concerned, the highest acceleration was found in the Y axis, followed by Z, with X showing least acceleration. The value was higher when taking into account Nrv. Data are shown in Table 2. 

Fatigue detection from acceleration was obtained by data from the Nrv of the three axes of movement. After signal preprocessing, fatigue was addressed by analyzing the relative energy computed from the DWT coefficients at levels 4 and 5. In this analysis, two steeper slopes were found in relative energy percentages from levels 4 and 5, (see Figure 5). The highest percentage of slope appeared at second 19, in which the percentage of relative energy from acceleration increased. This indicates the cut-off time point in which fatigue appears. This fatigue starting point is represented in Figure 5. 

## 4. Discussion

The aim of this study was to develop a detection system based on acceleration measured with a smartphone in order to analyze fatigue during a 30-STS test with EMG as the criterion. To our knowledge, this is the first study to detect fatigue by both acceleration and surface EMG. Fatigue was assessed as the difference detected in two variables of different nature. The main finding of the present study was that the two signals showed different cut-off points for fatigue detection, which did not correspond temporally. When the SC electromyograph was analyzed, the signal from the VM muscle showed fatigue firing at 12 seconds; however, analysis of the acceleration signal using the proposed system showed an increase in the percentage of relative energy indicating fatigue, appearing at 19 S.

As explained in the introduction, previous literature has shown that fatigue can be detected by observing a downward shift in the MNF or MFD. The present approach is based on a similar measure, namely spectral (SC), which measures the center of gravity of the power spectrum [52]. As shown in Figure 3, both measures can detect fatigue by a decrease in the MDF and SC. The method implemented in the current study is based on the computation of the first derivative of the SC corresponding to its variation slope. Hence, positive values of the slope indicate an upward shift of the SC, whereas negative slopes indicate a downward shift of the SC. Consequently, the point at which the slope changes its sign (from positive to negative) corresponds to fatigue onset. Detecting muscle fatigue at second 12 through EMG, considered the gold standard [20,21,22,23], ensured that the subject got fatigued during this task. Therefore, changes in acceleration after second 12 were guaranteed to occur due to fatigue.

The time difference between EMG and acceleration signals could be explained by several factors. On one hand, the differences between the studied variables may have contributed, as STS movement of the trunk is a reflection of the activity of the lower limbs [54]. Thus, there could be a correlation between the muscle fatigue of the quadriceps and trunk acceleration. However, muscular activity represents modifications in muscular membrane properties and metabolic conditions [24], whereas acceleration describes spatiotemporal characteristics [12]. On the other hand, the multidimensional nature of fatigue [25] could also explain the temporal differences in the detection of such a complex variable. In line with the complexity of fatigue, it is noteworthy that fatigue is defined as a transition stage in the electromyographic field, [55]. That is, there is a progressive process until fatigue appears [56]. It is well known that fatigue progresses with time and does not appear at an identifiable moment. In fact, metabolic muscle fatigue precedes contractible fatigue [57]. Changes in muscle fatigue have also been found to precede changes in joint kinematics [58]. Physiologically, the 30-STS test represents a short, high-intensity exercise in which adenosine triphosphate is mainly obtained from the sarcoplasmic stores of creatine phosphate, and then from glycolysis in the sarcoplasm. Changes in acceleration during this period may be evidenced by exhaustion of the metabolic reserves of creatine phosphate and accumulation of inorganic phosphate in the muscle tissue, giving way to anaerobic glycolysis and the accumulation of mainly lactic acid [47]. This is also related to changes in metabolic pathways, which represents another gradual process. Hence, any timed cut-off point given is a temporary approximation of mechanics that may change during fatigue.

Figure 3 shows the slope of the SC computed for each EMG channel, and the red dot indicates where the slope changes from positive to negative. Thus, it is possible to detect the starting point of fatigue. In this line, fatigue appears at second 12, as the SC signal from the VM muscle represents its highest frequency, followed by a descending slope of 12%. This happened because when fatigue from submaximal forces maintained is studied through surface EMG, the amplitude of its signal usually increases considerably due to the recruitment of extra motor units and an increase in firing frequency, followed by a decrease to optimize the stimulus rate with the prolonged duration of the muscle fibre action potential [59]. The shift to the lower frequencies is mainly explained as a diminished conduction along the sarcolemma and through central changes in motor unit firings [60].

Previous studies have focused on EMG and kinematics during this test. Regarding EMG, a previous study that focused on EMG during 30-STS found that VM was the muscle most likely to become fatigued [32], which is consistent with our results. Regarding kinematics, the term “biomechanical fatigue” was coined for fatigue obtained by comparing variation in joint angles execution before and after a fatiguing activity. However, fatigue was not obtained by the execution of the activity itself. As mentioned in the introduction, only one study has used velocity kinematics to study fatigability [42]. In the present study, acceleration can be considered a kinematic criterion itself, as the subject had to shift the maximum of its body mass as quickly as possible. The subject reached a point when they could no longer continue due to fatigue, which is significant from a clinical point of view and can be quantified.

Although RA, RF, SO, and GM were the muscles with higher MVC percentages (level of activation relative to its real activation value), the muscle whose fatigue was found to be related to SC variation was the VM. The importance of the VM muscle during STS has been previously studied, defending the position that knee extensors are among the main executors in this task [61,62], and that it is excessively loaded when muscle weakness develops under a loaded condition [17]. Furthermore, a previous study analyzing fatigue in different STS variants in the same muscles as the present study found that there were differences in fatigue from the VM muscle among variants, which was affected by the number of repetitions. Hence, the VM was the muscle more likely to become fatigued [32]. 

Regarding acceleration on each axis, the highest value was found on the Y axis, representing main acceleration during vertical motion. This result is in line with previous studies that have focused on vertical acceleration [16] and vertical velocity [15] during STS. 

As stated in the introduction, fatigue is a multidimensional concept, comprising both physiological and psychological factors [25]. Due to its complexity and the difficulty in isolating its different mechanisms, fatigue is difficult to define [63]. There are numerous factors that influence both STS and fatigue that may have limited the present study. The methodology and procedure selected also have their own strengths and limitations with regard to the fatigue challenge. Although the task was explained clearly and the subject was told to perform STS in the same starting position, it should be taken into account that muscle fatigue drives redistribution of the contribution of active muscles and postural adjustment [64], which could have influenced the performance of the task. Furthermore, each trial was carried out with a 1-week interval between each measure, and this long period of time may have also influenced fatigue. However, if the trials were performed on consecutive days without a rest period, this could have had a greater influence on fatigue. As the 30-STS test is a short, high-intensity exercise, the energy for muscle contraction is mainly obtained from sarcoplasmic stores of creatine phosphate [47]. A rest period of 1 week ensured that fatigue was not influenced by insufficient rest.

Another limitation is that the cut-off values identified for both time points are only applicable to one person. However, the measures were carried out in a single individual in order to avoid differences due to variables that influence fatigue, such as gender [65] and age [66], as well as variables that influence STS motion, such as BMI [67] and the individual’s self-perception and cognition [68]. Regarding BMI, it is known that obese subjects have a different motion strategy when performing sit-to-stand movements [67]. However, the obesity and sedentary lifestyle of the subject ensured that the task was fatigable enough, and that no other physical or sporting activity practiced by the subject could have influenced the results.

This study allowed us to detect fatigue through a smartphone placed on the participant’s sternum during the 30-STS test. This allowed fatigue to be quantified by the kinematic variable of acceleration, having been preceded by electromyography. As stated in the introduction, STS is an important patient outcome measure which is based mainly on the number of repetitions performed [8]. Although this variable is known to differ between populations, research has shown that a variable such us acceleration, which qualifies motion, provides valuable information and is more sensitive when discriminating between certain populations than the number of cycles completed [13]. A strength of the current study was that we used an acceleration signal to detect a variable as important as fatigue using the STS test, which is widely used. That is to say, while fatigue was previously only able to be acquired through EMG, this is the first time, to the authors’ knowledge, that fatigue has been quantified based on acceleration measured by a smartphone, a tool which is available to all clinicians.

Variables such as the number of repetitions and acceleration have allowed different populations to be distinguished in previous research; however, these populations are also expected to be distinguishable using a variable which as complex as fatigue and which has a multitude of factors that condition human movement. In line with this, the number of repetitions performed during 30-STS have been included as part of fatigue index in the cancer population [69]. Applying the system described here in a healthy sample would allow cut-off points to be obtained based on acceleration, which could then be compared with population, as well as establishing a stratification system [69] or elaborating classification tools based on fatigue in clinical populations [70]. This should be the focus of further research. In this study, we were able to detect fatigue by acceleration, representing the kinematic dimension of fatigue, as well as other physiological mechanic factors occurring simultaneously in the subject.

## 5. Conclusions

In this study, fatigue was detected by both trunk acceleration and surface EMG during a 30-STS test. The timed cut-off points determined for each variable were not temporally correlated, with fatigue being detected at 12 and 19 seconds for muscle fatigue localized in VM and trunk acceleration, respectively. The results of the present study will help clinicians obtain information about fatigue, a complex variable, through acceleration provided by an accessible and inexpensive device, i.e., a smartphone. Future research should apply the system developed in the present study to a sample consisting of varied individuals in order to determine a cut-off time for fatigue that is representative of different populations.

## Figures and Tables

**Figure 1 sensors-19-04202-f001:**
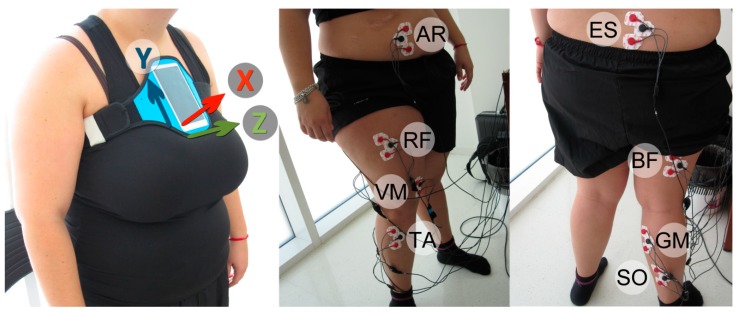
*X, Y* and *Z* axes and their direction from a Smartphone placed in Sternum (**left**) and EMG sensor placement (**middle** and **right**) in gastrocnemius medialis (GM), the biceps femoris (BF), the vastus medialis of the quadriceps (VM), the abdominal rectus (AR), the erector spinae (ES), the rectus femoris (RF), the soleus (SO), and the tibialis anterior (TA).

**Figure 2 sensors-19-04202-f002:**
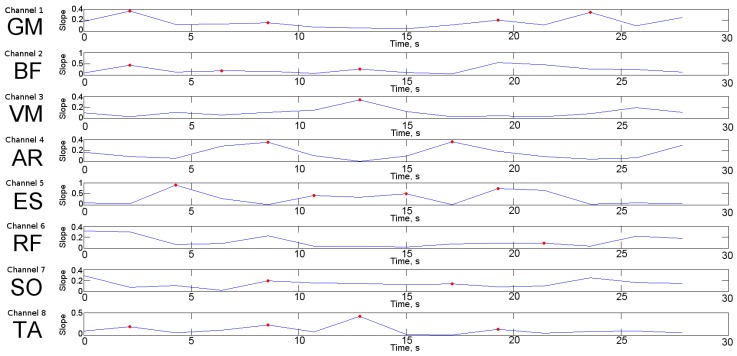
SC variation from EMG preprocessed and averaged signals from 8 measurements. Each channel represents the signal from the gastrocnemius medialis (GM), the biceps femoris (BF), the vastus medialis of the quadriceps (VM), the abdominal rectus (AR), the erector spinae (ES), the rectus femoris (RF), the soleus (SO), and the tibialis anterior (TA) muscles over a duration of 30 seconds. Slope indicates the spectrum shift (lower to higher or higher to lower frequencies) in each EMG channel. Red dots indicate the point where shifting to lower frequencies starts.

**Figure 3 sensors-19-04202-f003:**
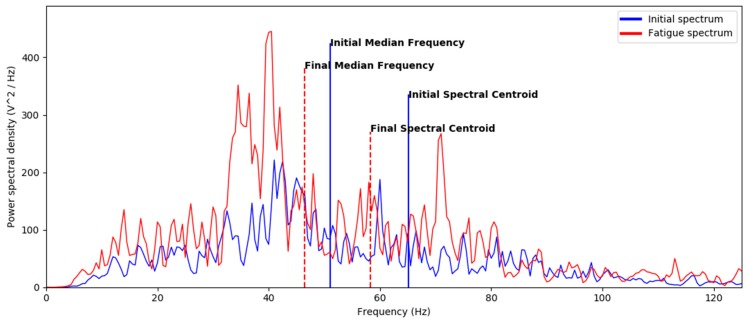
EMG power spectrum obtained for the initial time window (at the beginning of the 30-STS) and the final time window (the end of the 30-STS). As it is shown, a decrease in both the MDF and the SC occurs during the exercise.

**Figure 4 sensors-19-04202-f004:**
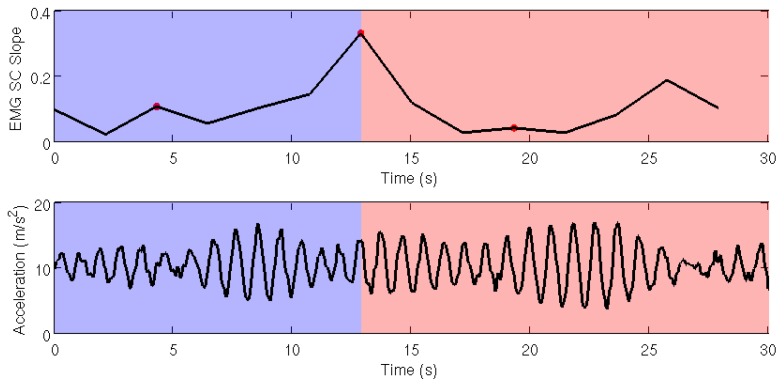
EMG SC slope from VM muscle signal (channel 3) and trunk acceleration during 30-STS. The cut-off time at second 12 corresponds to a descriptive acceleration value of 0.98 m/s^2^.

**Figure 5 sensors-19-04202-f005:**
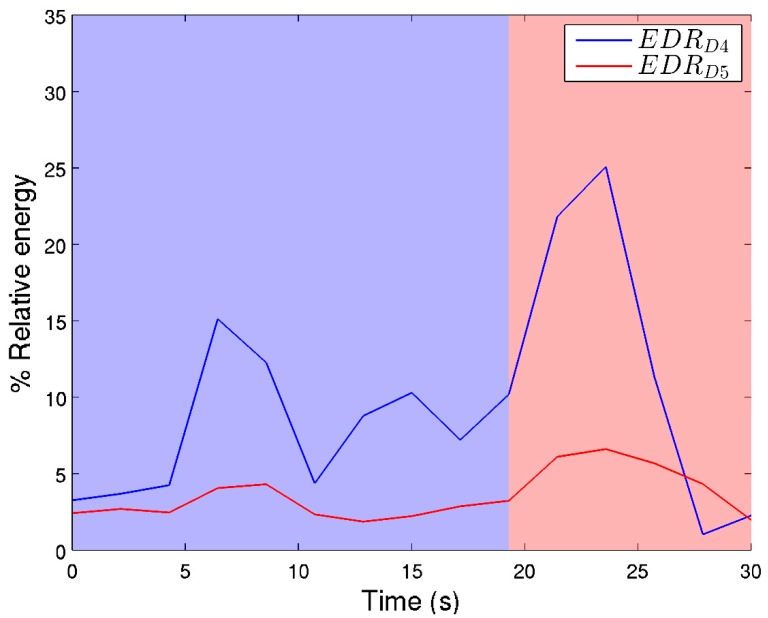
Fatigue detection from inertial data. The figure shows the energy of the subbands corresponding to the detailed DWT coefficients at levels 4 and 5. Red shaded part of the figure indicates the fatigue starting point at second 19.

**Table 1 sensors-19-04202-t001:** Electromyographic descriptive variables (mean, SD).

Muscles	Real Activation (µV)	Average Activation (µV)	MVC Percentage (%)	Muscle Involvement (%)
GM	288	128 (24.04)	42.88 (2.20)	8.5 (0.70)
BF	474	148 (41.01)	28.05 (4.47)	9.5 (2.12)
VM	762	212.50 (10.60)	27.46 (0.60)	14 (1.41)
AR	145	106.50 (10.60)	83.62 (14.38)	7 (1.41)
ES	81	116.50 (7.77)	14.80 (0.99)	7.5 (0.70)
RF	493	269 (5.65)	52.53 (2.86)	17.5 (2.12)
SO	492	263 (100.4)	49.08 (6.18)	15.5 (3.53)
TA	1098	298 (14.14)	28.64 (2.12)	21 (0.0)

**Table 2 sensors-19-04202-t002:** Acceleration (m/s^2^) in each axis and norm of the resultant vector (mean, SD).

Axis.	Mean (SD)
X	3.88 (0.39)
Y	25.79 (4.16)
Z	20.72 (0.28)
Nrv	28.04 (3.67)

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
