# Peer review of "Fatigue Detection during Sit-To-Stand Test Based on Surface Electromyography and Acceleration: A Case Study"

_sensors, 2019, doi:10.3390/s19194202_

Round 1

Reviewer 1 Report

This paper presents a fatigue detection system during the 30-second STS test using  acceleration and EMG data. The authors present their case study with one subject. Their goal is to detect when fatigue appears during the test, both in terms of muscle activity (EMG) and acceleration data. The manuscript includes useful information about sensor placement and data processing and analysis. However, there are a few parts of the paper that need improvement.

First of all, regarding paper organization, I suggest that most of the Discussion section should be moved to a Related Work section, since it includes very interesting information and background on fatigue and fatigue detection and will help the reader go through the rest of the paper easier. 

While the approach for fatigue detection using EMG data is much clearer and more elaborate, I found it hard to understand the acceleration-based fatigue detection approach. Since fatigue is a condition hard to define and detect, more references should be used to justify the proposed methods. Moreover, the nature of fatigue is highly subjective. I would like to see some references for this, as well as how self-report data (user input) could be used with this approach.

The authors state that they used on subject to avoid inter-subject variability. While this is a valid point, more subjects would make this study more reliable and interesting, even with user-specific analysis. I would also add a figure next to Figure 1 showing the EMG sensor placement, including all the muscles involved, eg., TA, SO, RF, etc.

Since it is hard to draw conclusions using only one subject, I believe that the paper has some interesting information and it could focus on the experimental setup, sensor placement and data analysis approach. The authors should improve the manuscript making such sections clearer and easier for the reader. The most basic improvement would be to clarify and justify why fatigue is detected in such a way using these two modalities.

Author Response

Reviewer 1:

This paper presents a fatigue detection system during the 30-second STS test using  acceleration and EMG data. The authors present their case study with one subject. Their goal is to detect when fatigue appears during the test, both in terms of muscle activity (EMG) and acceleration data. The manuscript includes useful information about sensor placement and data processing and analysis. However, there are a few parts of the paper that need improvement. 

First of all, regarding paper organization, I suggest that most of the Discussion section should be moved to a Related Work section, since it includes very interesting information and background on fatigue and fatigue detection and will help the reader go through the rest of the paper easier

Authors: Thank you. We do agree with you, as mostly parts of discussion fits better in introduction section. However, they are not extensive enough for a specific “related work” section. Therefore, it has been included in introduction section as follows:

In the peripheral system, muscle, nerves and glycogen stores comprise physiologic factors [26]. Muscle fatigue study can be conducted by measuring EMG signals using surface electrodes to search for specific patterns. However, information related to fatigue is not present in the signal amplitude but in the frequency spectrum. In fact, the most popular methods for fatigue detection using EMG signals are based on measurement of the mean frequency and median frequency in the signal spectrum [28–31]. Specifically, [28–30] show that muscle fatigue results in a downward shift of the frequency content of EMG signals. As a result, fatigue can be detected by observing a downward shift in the MNF or MDF values” (lines 69-73)

New references:

Kluger, B.M.; Krupp, L.B.; Enoka, R.M. Fatigue and fatigability in neurologic illnesses: proposal for a unified taxonomy. Neurology 2013, 80, 409–416.

Related to the present work, fatigue during the 30-STS test has been previously analyzed in the lower limbs and trunk muscles by EMG [32]. Muscle fatigue is defined as a state of depressed responsiveness resulting from activity [33]. Hence it constitutes a very important parameter providing motion quality information because it affects kinematic parameters in the lower limbs [34–38]. Regarding kinematics, acceleration during STS has been related to variables such as peak power [39].  Previous studies have also highlighted impairment of movement due to fatigue. This even includes what is known as “biomechanical fatigue”, referring to fatigue that is measured by speed and acceleration, and is closely related to kinematics [22]In this field, variables such as velocity [40] or acceleration [41] can differentiate between fatigued and non-fatigued training exercises. However, these kinematic differences were analyzed by comparing execution before and after a fatiguing activity, and the variation in joint angles was used as an indicator of fatigue rather than the execution itself. Only one previous study considered a reduction in the percentage of trunk velocity with respect to the maximum value recorded during the first repetitions of STS. This study assessed fatigability in older women using a linear encoder fixed with a belt on the hip, which measured the distance per time unit [42]”. (lines 78-92)

In order to keep a common thread among discussion section, the information given was summarized, as follows:

As explained in the introduction, previous literature has shown that fatigue can be detected by observing a downward shift in the MNF or MFD. The present approach is based on a similar measure, namely spectral (SC), which measures the center of gravity of the power spectrum. Moreover, the method implemented in the current study is based on computation of the first derivative of the SC corresponding to its variation slope. Hence, positive values of the slope indicate an upward shift of the SC whereas negative slopes indicate a downward shift of the SC. Consequently, the point at which the slope changes its sign (from positive to negative) indicates fatigue onset. Detecting muscle fatigue at second 12 through EMG, considered the gold standard [20–23], (SC),(lines 265-273)

New references:

Bonato, P.; Roy, S.H.; Knaflitz, M.; De Luca, C.J. Time-frequency parameters of the surface myoelectric signal for assessing muscle fatigue during cyclic dynamic contractions. IEEE Trans Biomed Eng 2001, 48, 745–753. Farina, D.; Pozzo, M.; Merlo, E.; Bottin, A.; Merletti, R. Assessment of average muscle fiber conduction velocity from surface EMG signals during fatiguing dynamic contractions. IEEE Trans Biomed Eng 2004, 51, 1383–1393 Wu, Q.; Mao, J.F.; Wei, C.F.; Fu, S.; Law, R.; Ding, L.; Yu, B.T.; Jia, B.; Yang, C.H. Hybrid BF–PSO and fuzzy support vector machine for diagnosis of fatigue status using EMG signal features. Neurocomputing 2016, 173, 483–500.

Previous studies have focused on EMG and kinematics during this test. Regarding EMG, a previous study that focused on EMG during 30-STS found that VM was the muscle most likely to become fatigued [32], which is consistent with our results. Regarding kinematics, the term “biomechanical fatigue” was coined for fatigue obtained by comparing variation in joint angles execution before and after a fatiguing activity. However, fatigue was not obtained by the execution of the activity itself. As mentioned in the introduction, only one study has used velocity kinematics to study fatigability [48]. In the present study,” (lines 303-309, page 9)

While the approach for fatigue detection using EMG data is much clearer and more elaborate, I found it hard to understand the acceleration-based fatigue detection approach. Since fatigue is a condition hard to define and detect, more references should be used to justify the proposed methods. Moreover, the nature of fatigue is highly subjective. I would like to see some references for this, as well as how self-report data (user input) could be used with this approach. 

Authors: Thank you. We do agree with you in the complexity of fatigue. Therefore, we have added new information in introduction section in order to define its complexity, dimensions and multiple factors.  This information is complemented by the subjective dimension and new references. This part can be read as follows:

“One of the most studied variables by EMG is fatigue, as EMG is considered the gold-standard for measuring fatigue [20–23]. However, changes in the EMG signal represents modifications in muscular membrane properties and metabolic conditions [24], while fatigue is a multidimensional concept that involves psychological and physiological dimensions [25]: On the one hand, the psychological or subjective dimension of fatigue refers to perceptions of fatigue, and is composed by psychological factors such as perceptions of effort, expectations, motivation, arousal or mood [26]. This subjective dimension is usually measured by patient-reported outcomes [27]. On the other hand, the physiological dimension of fatigue can be observed in both the central and peripheral system domains [25]. In the peripheral system, muscle, nerves and glycogen stores comprise physiologic factors [26]. Muscle fatigue study can be conducted by measuring EMG signals using surface electrodes to search for specific patterns.” (lines 61-71).

New references:

Bonato, P.; Roy, S.H.; Knaflitz, M.; De Luca, C.J. Time-frequency parameters of the surface myoelectric signal for assessing muscle fatigue during cyclic dynamic contractions. IEEE Trans Biomed Eng 2001, 48, 745–753. Farina, D.; Pozzo, M.; Merlo, E.; Bottin, A.; Merletti, R. Assessment of average muscle fiber conduction velocity from surface EMG signals during fatiguing dynamic contractions. IEEE Trans Biomed Eng 2004, 51, 1383–1393 Wu, Q.; Mao, J.F.; Wei, C.F.; Fu, S.; Law, R.; Ding, L.; Yu, B.T.; Jia, B.; Yang, C.H. Hybrid BF–PSO and fuzzy support vector machine for diagnosis of fatigue status using EMG signal features. Neurocomputing 2016, 173, 483–500. Kluger, B.M.; Krupp, L.B.; Enoka, R.M. Fatigue and fatigability in neurologic illnesses: proposal for a unified taxonomy. Neurology 2013, 80, 409–416. Hewlett, S.; Dures, E.; Almeida, C. Measures of fatigue: Bristol Rheumatoid Arthritis Fatigue Multi-Dimensional Questionnaire (BRAF MDQ), Bristol Rheumatoid Arthritis Fatigue Numerical Rating Scales (BRAF NRS) for severity, effect, and coping, Chalder Fatigue Questionnaire (CFQ), Checklist Individual Strength (CIS20R and CIS8R), Fatigue Severity Scale (FSS), Functional Assessment Chronic Illness Therapy (Fatigue) (FACIT-F), Multi-Dimensional Assessment of Fatigue (MAF), Multi-Dimensional Fatigue Inventory (MFI), Pediatric Quality Of Life (PedsQL) Multi-Dimensional Fatigue Scale, Profile of Fatigue (ProF), Short Form 36 Vitality Subscale (SF-36 VT), and Visual Analog Scales (VAS). Arthritis Care Res (Hoboken) 2011, 63 Suppl 11, S263-286.

At a physiological and peripheral domain, muscle fatigue is detected by EMG. Therefore, the prior paragraph is followed by information regarding fatigue detection by EMG. In that paragraph, we explain that median frequency

This information is followed by information regarding fatigue detection by EMG:

However, information related to fatigue is not present in the signal amplitude but in the frequency spectrum. In fact, the most popular methods for fatigue detection using EMG signals are based on measurement of the mean frequency and median frequency in the signal spectrum [28–31]. Specifically, [28–30] show that muscle fatigue results in a downward shift of the frequency content of EMG signals. As a result, fatigue can be detected by observing a downward shift in the MNF or MDF values” (lines 71-76).

Therefore, the methodology employed allowed us to obtain muscle fatigue, considered the gold standard (first paragraph) though the methodology employed in the literature (second paragraph). Detecting fatigue at second 12 though EMG  ensured us the subject reached fatigue in this test, therefore, changes in acceleration after this temporally cut-off would guarantee are due to fatigue. Authors have attempted to clarify this issue by modifying discussion section: Although the first paragraph from discussion section remains the same, it is followed by an explanation of the methodology employed by EMG, the implications of obtaining fatigue by EMG in acceleration findings, followed by an explanation in time differences between both signals. This can be read as follows:

The aim of this study was to develop a detection system based on acceleration measured with a smartphone in order to analyze fatigue during the 30-STS test with EMG as the criterion. To our knowledge, this is the first study to detect fatigue by both acceleration and surface EMG. Fatigue was assessed as the difference detected in two variables of different nature. The main finding of the present study was that the two signals showed different cut-off points for fatigue detection, which did not correspond temporally. When the SC electromyograph was analyzed, the signal from the VM muscle showed fatigue firing at 12 seconds; however, analysis of the acceleration signal using the proposed system showed an increase in the percentage of relative energy indicating fatigue, appearing at 19 seconds.

As explained in the introduction, previous literature has shown that fatigue can be detected by observing a downward shift in the MNF or MFD. The present approach is based on a similar measure, namely spectral (SC), which measures the center of gravity of the power spectrum. Moreover, the method implemented in the current study is based on computation of the first derivative of the SC corresponding to its variation slope. Hence, positive values of the slope indicate an upward shift of the SC whereas negative slopes indicate a downward shift of the SC. Consequently, the point at which the slope changes its sign (from positive to negative) indicates fatigue onset. Detecting muscle fatigue at second 12 through EMG, considered the gold standard [20–23], assured that the subject got fatigued during this task. Therefore, changes in acceleration after the second 12 guaranteed to occur due to fatigue.       

This time difference between EMG and acceleration signals could be explained by several factors. On one hand, the differences between the studied variables may have contributed, as STS movement of the trunk is a reflection of the activity of the lower limbs” (lines 256-277).

New references:

Bonato, P.; Roy, S.H.; Knaflitz, M.; De Luca, C.J. Time-frequency parameters of the surface myoelectric signal for assessing muscle fatigue during cyclic dynamic contractions. IEEE Trans Biomed Eng 2001, 48, 745–753. Farina, D.; Pozzo, M.; Merlo, E.; Bottin, A.; Merletti, R. Assessment of average muscle fiber conduction velocity from surface EMG signals during fatiguing dynamic contractions. IEEE Trans Biomed Eng 2004, 51, 1383–1393 Wu, Q.; Mao, J.F.; Wei, C.F.; Fu, S.; Law, R.; Ding, L.; Yu, B.T.; Jia, B.; Yang, C.H. Hybrid BF–PSO and fuzzy support vector machine for diagnosis of fatigue status using EMG signal features. Neurocomputing 2016, 173, 483–500….

The authors state that they used on subject to avoid inter-subject variability. While this is a valid point, more subjects would make this study more reliable and interesting, even with user-specific analysis. I would also add a figure next to Figure 1 showing the EMG sensor placement, including all the muscles involved, eg., TA, SO, RF, etc. 

Since it is hard to draw conclusions using only one subject, I believe that the paper has some interesting information and it could focus on the experimental setup, sensor placement and data analysis approach. The authors should improve the manuscript making such sections clearer and easier for the reader. The most basic improvement would be to clarify and justify why fatigue is detected in such a way using these two modalities. 

Authors: Thank you. Due to your comments about figures, experimental setup and sensor placement, figure 2 (test starting position) was replaced by a new figure showing EMG sensor placement (new figure 1). Furthermore, for better comprehension of procedure, a video as supplementary material has been included, so readers can see 30-STS performance. Authors hope this helps to clarify the present manuscript.  

Regarding the use of these two modalities for detecting fatigue, this comment is closely related to the previous one. The justification and clarification of how can fatigue be detected by this those modalities were explained in the previous comment, with refers to introduction section (lines lines 71-76) and discussion section (lines 256-277).

Regarding acceleration, analyzing fatigue through energy percentage is based on reference #52. When looking at the energy percentage of levels 4 and 5, two steeper slopes are produced. Its percentage of slope varies very little, but as the one that occurs in the second 19 has a higher value, it was the one chosen. It should be taken into account that the present study aims to develop a fatigue detection system based on acceleration analyzing EMG as reference. Hence, it is a criterion that authors have chosen, since nothing similar has been done before. The modality provided implies a novelty. As mentioned in discussion section, only one previous study considered a reduction in the percentage of trunk velocity with respect to the maximum to asses fatigability.

The fatigue detection though acceleration is explained in methods sections:

“Inertial data processing is performed in the frequency domain, and consists of computing the average power on different subbands by means of the Discrete Wavelet Transform, computed in overlapped windows (overlapping factor = 0.5). Specifically, we computed the variance of the approximated and detailed coefficients from five levels using the Daubechies db4 as the mother wavelet. In addition, the calculation of the energy ratio of the approximated coefficients Ai and detailed coefficients Di helps to find a more clear pattern [52]. At level i, these are computed as EDRAi=AiAiT/ET and EDRDj=DiDjT/ET (j=1,...,i) for the approximation and detailed coefficients, respectively, where

[49]

Fatigue detection from inertial data has been addressed by analyzing the relative energy for each window as computed from the DWT coefficients at different levels. This way, a breakdown point in the average power can be computed from Level 4 and 5 coefficients. “ (lines 206-216)

Reference 52:

Ayrulu-Erdem, B.; Barshan, B. Leg motion classification with artificial neural networks using wavelet-based features of gyroscope signals. Sensors (Basel) 2011, 11, 1721–1743.

To clarify acceleration procedure, we added more information in results section, as follows:

Fatigue detection from acceleration was obtained by data from the Nrv of the three axes of movement. After signal pre-processing, fatigue was addressed by analyzing the relative energy computed from the DWT coefficients at levels 4 and 5. In this analysis, two steeper slopes were found in relative energy percentages from levels 4 and 5, (see figure 5). The highest percentage of slope appeared at second 19, in which the percentage of relative energy from acceleration increased. This indicates the cut-off time point in which fatigue appears. This fatigue starting point is represented in figure 5. “(lines  250-256)

Reviewer 2 Report

This case study presents a method to detect muscle fatigue using accelerometry, using surface EMG as the criterion measure.

The authors present a method to detect onset of fatigue using non-standard EMG measures - median frequency should be added as a reference measure. Results are presented based on 8 repetitions of a 30 s sit-to-stand test, however it is difficult to interpret the presented results as many details are missing, and figures are not clear. 

This reviewer would suggest adding more subjects, and a standard measure of muscle fatigue, and to improve the figures presented. 

Page 2, Line 57:  Qualitative should be Quantitative here I think.

Page 3, Line 94:  "side was recorded at a frequency of 10.000Hz." - was should be were... and 10.000Hz reads as 10Hz to me, should this be 10 kHz?  or 1000 Hz?

Line 116:   "The reliability and validity of accelerometer embedded in a Smartphone placed in sternum have 115 been previously provided [34]."  

Line 118: 25 Hz seems too low for sampling accelerometry data - 100 Hz would be recommended usually. Why did you choose this rate?

Figure 3: I would recommend making muscle names clearer in the figure. In the caption, medial gastrocnemius (MG) is used, whereas GM is  used in the Figure. "Represented along the time" does not make sense... 'Slope sign' is confusing also, the data looks all positive to me? So it is a value rather than a sign?  Finally - is this data from one test? If so, this should be stated.

Figure 4: This looks like representative data again, and should be stated in the caption. Figure 3 and Figure 4 seem repetitive.

Figure 5: Trials should be numbered... This figure does not tell us a lot. 

Line 204: 'The area under the curve for 204 the VM muscle signal decreases at second 12 with a slope of 12%.'

Line 226: 'The percentage of relative energy from acceleration showed an increase at second 19.' - surely there should be a mean and standard deviation on this time results?  As it should be averaged across all trials? Some sort of accuracy measure on this result is needed. 

Page 10: line 320 - missing 'on'

A rest period of one week was provided - did you weigh the subject each week? Her BMI may vary slightly from week to week...

Page 11, line 331: "This allowed fatigue to be quantified by the physiological variable of acceleration" - I would not classify acceleration as a physiological variable... kinematic?

Conclusion:

"Future research should apply the system developed in in the present study to sample consisting of varied individuals in order to determine a cut-off time for fatigue that is representative of different populations."

'in in' is a typo, and it should be 'a sample'

Finally - this future study is the one that I think warrants publication.  One subject is not enough here...

Summary: My impression of this paper is mixed, it seems like some good work was conducted, but the presentation of results needs work. Even with one subject (which needs to be increased) you have not clearly presented a new method to detect fatigue using accelerometry.  Improved analysis methods, presentation of results and larger subject numbers are required. I will be interested to read the final paper, but I feel there is still work to be done.  

Author Response

Reviewer 2:

Comments and Suggestions for Authors

This case study presents a method to detect muscle fatigue using accelerometry, using surface EMG as the criterion measure.

The authors present a method to detect onset of fatigue using non-standard EMG measures - median frequency should be added as a reference measure. Results are presented based on 8 repetitions of a 30 s sit-to-stand test, however it is difficult to interpret the presented results as many details are missing, and figures are not clear. 

This reviewer would suggest adding more subjects, and a standard measure of muscle fatigue, and to improve the figures presented. 

Authors: Thank you. As we detailed in the manuscript “Classical EMG processing used to identify muscle fatigue using surface electrodes is based on the detection of a decrease in the median frequency (MF) of the EMG spectrum [48–50] during an isometric muscle action”. (lines 186-188).This paragraph has been reinforced by a new references:.

Allison, G.T.; Fujiwara, T. The relationship between EMG median frequency and low frequency band amplitude changes at different levels of muscle capacity. Clin Biomech (Bristol, Avon) 2002, 17, 464–469. De Luca, C.J.; Sabbahi, M.A.; Roy, S.H. Median frequency of the myoelectric signal. Effects of hand dominance. Eur J Appl Physiol Occup Physiol 1986, 55, 457–464.

Therefore, median frequency is considered as a reference measure for muscle fatigue in the literature. However, fatigue analysis can also be done by our approach based on spectral centroid, and this is reflected in previous literature. For better comprehension, we clarified this information with a new reference. The previous mentioned paragraph is followed by the next one:

 “In fact, MF and spectral centroid (SC) are both considered measures of central tendency of the spectral distribution [51].

New Reference.         51.       Chiang, S.; Vankov, E.R.; Yeh, H.J.; Guindani, M.; Vannucci, M.; Haneef, Z.; Stern, J.M. Temporal and spectral characteristics of dynamic functional connectivity between resting-state networks reveal information beyond static connectivity. PLoS One 2018, 13.

This explanation has been reinforced in the discussion section, as follows:

As explained in the introduction, previous literature has shown that fatigue can be detected by observing a downward shift in the MNF or MFD. The present approach is based on a similar measure, namely spectral (SC), which measures the center of gravity of the power spectrum [51]. Moreover, the method implemented in the current study is based on computation of the first derivative of the SC corresponding to its variation slope. Hence, positive values of the slope indicate an upward shift of the SC whereas negative slopes indicate a downward shift of the SC. Consequently, the point at which the slope changes its sign (from positive to negative) indicates fatigue onset. Detecting muscle fatigue at second 12 through EMG, considered the gold standard [20–23], assured that the subject got fatigued during this task. Therefore, changes in acceleration after the second 12 guaranteed to occur due to fatigue“ (lines 271-280).

Regarding figures, in response to comment #3 from reviewer 1, we explain all the changes made, including a new figure 2 and modifying figure 3, which concur with some editor´s suggestion. Furthermore, a new video has been included as supplementary file.

Furthermore, we have included information in the last part from introduction section (related work suggested by reviewer 1), preprocessing (lines 174-176;179-180), data processing (188-189, new references 50 and 51), results (figure 3 explanation, and fatigue from acceleration in lines 251-256) and discussion section.

After all details given, authors hope results are easier to interpret.

Page 2, Line 57:  Qualitative should be Quantitative here I think.

Authors: Thank you. The term “qualitative” refers to “how the movements were carried” (line 54) in others terms different from number of repetitions (quantitative). This can be better understand by reading previous lines “Lately, several studies have shown that besides quantifying motion, it is important to provide quality information about how the motion is carried out through kinematic parameters” (lines 45-47).

Page 3, Line 94:  "side was recorded at a frequency of 10.000Hz." - was should be were... and 10.000Hz reads as 10Hz to me, should this be 10 kHz?  or 1000 Hz?

Authors: Thank you for this comment. It was a typing error. It has been replaced by 1.000Hz

Line 116:   "The reliability and validity of accelerometer embedded in a Smartphone placed in sternum have 115 been previously provided [34]."  

Authors: Thank you. We have eliminated capital letter from smartphone.

Line 118: 25 Hz seems too low for sampling accelerometry data - 100 Hz would be recommended usually. Why did you choose this rate?

Authors: Thank you. Inertial sensors devices usually employs a frecuency of 100Hz, like studies with Xsens [16] or inertiacube [34]. However, latest studies has shown that lower frequencies obtained by Smartphones (for example, 32 Hz) are also reliable [34]. This allow to transfer information to cheaper and and easy-to-use devices such as smartphone. In this case, both the Smartphone and the app employed allow recording acceleration data at 25Hz.

Millor, N.; Lecumberri, P.; Gomez, M.; Martinez-Ramirez, A.; Rodriguez-Manas, L.; Garcia-Garcia, F.J.; Izquierdo, M. Automatic evaluation of the 30-s chair stand test using inertial/magnetic-based technology in an older prefrail population. IEEE J Biomed Health Inform 2013, 17, 820–827 Galán-Mercant, A.; Barón-López, F.J.; Labajos-Manzanares, M.T.; Cuesta-Vargas, A.I. Reliability and criterion-related validity with a smartphone used in timed-up-and-go test. Biomed Eng Online 2014, 13, 156.

Figure 3: I would recommend making muscle names clearer in the figure. In the caption, medial gastrocnemius (MG) is used, whereas GM is used in the Figure. "Represented along the time" does not make sense... 'Slope sign' is confusing also, the data looks all positive to me? So it is a value rather than a sign?  Finally - is this data from one test? If so, this should be stated.

Authors: Thank you so much for this comment. We have replaced “medial gastrocnemius”(MG) by “gastrocnemius medialis”(GM) along the whole manuscript for consistence. Furthermore, we have included bigger words in figure 3 for muscles names.  Regarding description, “along the time” has been replaced by “along the 20 seconds” (we meant the duration of the 30-STS test”. We do agree with you about slope sign: we meant higher or lower frequencies, but in this case, all values are positive. Therefore “sign” has been eliminated from that paragraph.

The data represent the average from 8 trials, as explained in methods section. This has been stated and clarified in that paragraph, as follows: “Each channel represent SC from averaged 8 measurements after being rectified, filtered and temporarily aligned” (lines 229-230)

Figure 4: This looks like representative data again, and should be stated in the caption.

Authors: Thank you. We have added in the caption the representative data relative to this figure and that was missing: “The cut-off time at second 12 corresponds to a descriptive acceleration value of 0.98 m/s2

Figure 3 and Figure 4 seem repetitive.

Authors: Thank you. In this new version of the manuscript, we have added more information regarding figure 3, as follows:

Each channel represent SC from averaged 8 measurements after being rectified, filtered and temporarily aligned”(lines 229-230).

Along with information added in caption from figure 4, we hope it gets easier to understand that figure 3 correspond to EMG SC slope in all 8 channels (8 muscles), from which channel 3 correspond to vastus medialis of the quadriceps, the one represented in figure 4 together with descriptive values of acceleration.

Figure 5: Trials should be numbered... This figure does not tell us a lot

Authors: Thank you. Due to this comment, this figure has been eliminated. Instead, as suggested by reviewer 1, we have included a new figure (Figure 2) showing the EMG sensor placement.

Line 204: 'The area under the curve for 204 the VM muscle signal decreases at second 12 with a slope of 12%.'

Authors: Thank you. Thank you. In the new version of the manuscript, that line is preceded by “In channel 3, which correspond to the VM muscle, a highlighted frequency shift between seconds 10 and 15 can be observed” (lines 232-233).

Line 226: 'The percentage of relative energy from acceleration showed an increase at second 19.' - surely there should be a mean and standard deviation on this time results?  As it should be averaged across all trials? Some sort of accuracy measure on this result is needed.

Authors: Thank you so much for this question. Data from the percentage of relative energy from acceleration and its representation in figure 6 are calculated from temporal averaged signal (as can be read in preprocessing section: “Once EMG and acceleration signals were rectified and filtered, signals obtained during eight repetitions of the experiment were averaged in order to search for a statistically significant pattern. This requires the temporal alignment of signals acquired during the 8 different experiments, which was addressed by taking the maximum value of cross-correlation with respect to the first acquisition as a reference. This way, further processing was carried out using the averaged signals” ,lines 179-184). Therefore, mean and standard deviation could be calculated in the time-domain graph, but not in the frequency spectrum, as done in acceleration data.

Page 10: line 320 - missing 'on'

Authors: Thank you. We have fixed it.

A rest period of one week was provided - did you weigh the subject each week? Her BMI may vary slightly from week to week...

Authors: Thank you. We do agree that BMI could have varied slightly. However, this variation is not likely to be significant. Cross-sectional studies usually do not report descriptive and anthropometric variables along weeks unless those variables were target variables (i.e expected weight lost), which is not the case of the present study. Furthermore, the subject was told to continue her normal activities of daily living. 

Page 11, line 331: "This allowed fatigue to be quantified by the physiological variable of acceleration" - I would not classify acceleration as a physiological variable... kinematic?

Authors: Thank you. We have fixed it.

Conclusion:

"Future research should apply the system developed in in the present study to sample consisting of varied individuals in order to determine a cut-off time for fatigue that is representative of different populations."

'in in' is a typo, and it should be 'a sample'

Authors: Thank you so much. We have fixed it.

Finally - this future study is the one that I think warrants publication.  One subject is not enough here...

Summary: My impression of this paper is mixed, it seems like some good work was conducted, but the presentation of results needs work. Even with one subject (which needs to be increased) you have not clearly presented a new method to detect fatigue using accelerometry.  Improved analysis methods, presentation of results and larger subject numbers are required. I will be interested to read the final paper, but I feel there is still work to be done.  

 Authors: Thank you. Regarding the inclusion of only one single subject, this is already contemplated as a limitation (“Another limitation is that the cut-off values identified for both time points are only applicable to one person”. lines 345-346). However, we also explained the advantages that drove authors to make this decision: However, the measures were carried out in a single individual in order to avoid differences due to variables that influence fatigue, such as gender [64] and age [65], as well as variables that influence STS motion, such as BMI [66] and the individual’s self-perception and cognition [67]”, lines 346-348). That is to say, each subject has its own movement pattern. So having 8 different subjects, even with the same gender, age and physical status, would have involved 8 different movement patterns.  However, measurements made in the same subject facilitates obtaining a more consistent pattern of movement (with intra-subject variability but without inter-subject variability) when creating a cut-off point and developing a method of obtaining fatigue. From our point of view, measurements with intra-subject variability are the previous step before applying it in measurements with inter-subject variability. As stayed in the manuscript, further research is need after this manuscript, but authors thought testing measures in one single subject was need prior testing the created methods to a wider sample.

As explained, the eight trials were carried out in one single subject to avoid inter-subject variability; and trials were average to create a unique pattern from a subject in order to decrease or eliminate within subject variability. Reporting subject variability and describing its pattern was not the goal, but creating a pattern as consistent as possible in both EMG and acceleration signal prior analyzing a cut-off point related to fatigue. To strength our methodology, some data regarding intra-subject variability is provided. Specifically, SD and SEM from measurements of the variation of the slope of the spectral centroid can be found in results section “Regarding intra-subject variability, the SEM and SD from the 8 averaged measures from VM were 0.05 and 0.15, respectively” (lines 231-232). Hence, although results from 8 test were averaged, intra-subject variability complement data.

As far as acceleration methodology concerns, this issue has been explained in response to the last comment from reviewer #1. In the tracked version of the manuscript, changed made in methods, results and discussion to clarity this method can be easily found.

Reviewer 3 Report

1 iIn order to identify muscle fatigue in EMG in a dynamic condition, the
spectral centroid (SC) (which is related to the “center of mass” of the discrete Fourier transform
(DFT) spectrum) was computed in each overlapped (0.5 overlapping factor) Hanning window.

please present the reason that spectral centroid can represent the fatigue under a dynamic condition.

2. More comparisons need clear the contribution of this work such as fatigue indexes and signals process methods.

3. More references need been cited. For example,

1) Qi Wu, Jianfeng Mao, C.F Wei, Shan Fu, Rob Law, L. Ding, B.T. Yu, B. Jia, C.H. Yang, " Hybrid BF-PSO and fuzzy support vector machine for diagnosis of fatigue status using EMG signal features ", Neurocomputing, 173, 483-500, 2016.

Author Response

Reviewer 3:

Comments and Suggestions for Authors

1 iIn order to identify muscle fatigue in EMG in a dynamic condition, the
spectral centroid (SC) (which is related to the “center of mass” of the discrete Fourier transform (
DFT) spectrum) was computed in each overlapped (0.5 overlapping factor) Hanning window. please present the reason that spectral centroid can represent the fatigue under a dynamic condition.

Authors: Thank you. This comment is closely related to comments done by reviewers 2 and 3.   The EMG emits pulses of electrical signal with a certain frequency. From the same electrode, many different frequencies come out, although there is always a predominant or central one. When passing from the pulse / time to pulse / frequency spectrum, when the center of the curve shifts should be detected. That center of the curve that moves would be the mentioned spectral centroid (SC).Both median frequency and SC can be used to study fatigue under dynamic conditions. This has been better explained in the manuscript, as follows:

Introduction section

Muscle fatigue study can be conducted by measuring EMG signals using surface electrodes to search for specific patterns. However, information related to fatigue is not present in the signal amplitude but in the frequency spectrum. In fact, the most popular methods for fatigue detection using EMG signals are based on measurement of the mean frequency and median frequency in the signal spectrum [28–31]. Specifically, [28–30] show that muscle fatigue results in a downward shift of the frequency content of EMG signals. As a result, fatigue can be detected by observing a downward shift in the MNF or MDF values.”  (lines 70-76)

Data processing, methods section

Classical EMG processing used to identify muscle fatigue using surface electrodes is based on the detection of a decrease in the median frequency (MF) of the EMG spectrum [48–50] during an isometric muscle action. In fact, MF and spectral centroid (SC) are both considered measures of central tendency of the spectral distribution[51]” (lines 186-189)

Discusion section

As explained in the introduction, previous literature has shown that fatigue can be detected by observing a downward shift in the MNF or MFD. The present approach is based on a similar measure, namely spectral (SC), which measures the center of gravity of the power spectrum[51]. Moreover, the method implemented in the current study is based on computation of the first derivative of the SC corresponding to its variation slope. Hence, positive values of the slope indicate an upward shift of the SC whereas negative slopes indicate a downward shift of the SC. Consequently, the point at which the slope changes its sign (from positive to negative) indicates fatigue onset. Detecting muscle fatigue at second 12 through EMG, considered the gold standard [20–23], assured that the subject got fatigued during this task. Therefore, changes in acceleration after the second 12 guaranteed to occur due to fatigue. “ (lines 273-282)

More comparisons need clear the contribution of this work such as fatigue indexes and signals process methods.

Authors: Thank you.  This has been into account in the last paragraph from discussion section:

…” In line with this, the number of repetitions performed during 30-STS have been included as part of fatigue index in cancer population [68]. Applying the system described here in a healthy sample would allow cut-off points to be obtained based on acceleration, which could then be compared with population, as well as stablish stratification system [68] or elaborate classification tools based on fatigue in clinical populations [69].”(lines 368-372).

New References:

Cuesta-Vargas, A.; Buchan, J.; Pajares, B.; Alba, E.; Roldan-Jiménez, C. Cancer-related fatigue stratification system based on patient-reported outcomes and objective outcomes: A cancer-related fatigue ambulatory index. PLOS ONE 2019, 14, e0215662. Cuesta-Vargas, A.; Luciano, J.V.; Peñarrubia-María, M.T.; García-Campayo, J.; Fernández-Vergel, R.; Arroyo-Morales, M.; Serrano-Blanco, A.; FibroQoL Study Group Clinical dimensions of fibromyalgia symptoms and development of a combined index of severity: the CODI index. Qual Life Res 2013, 22, 153–160 More references need been cited. For example,

1) Qi Wu, Jianfeng Mao, C.F Wei, Shan Fu, Rob Law, L. Ding, B.T. Yu, B. Jia, C.H. Yang, " Hybrid BF-PSO and fuzzy support vector machine for diagnosis of fatigue status using EMG signal features ", Neurocomputing, 173, 483-500, 2016.

Authors: Thank you.  That reference has been used to reinforce the use of EMG to measure muscle fatigue, as follows:

One of the most studied variables by EMG is fatigue, as EMG is considered the gold-standard for measuring fatigue [20–23].”(introduction, lines 61-62).

New reference:

Wu, Q.; Mao, J.F.; Wei, C.F.; Fu, S.; Law, R.; Ding, L.; Yu, B.T.; Jia, B.; Yang, C.H. Hybrid BF–PSO and fuzzy support vector machine for diagnosis of fatigue status using EMG signal features. Neurocomputing 2016, 173, 483–500.

All new references can be observed by tracked version of the manuscript, which correspond to references 20, 21, 23, 26, 27, 49, 50, 51, 68 and 69.

Round 2

Reviewer 2 Report

Many of my concerns from the previous review still stand. 

You have not adequately justified your criterion measure – I still recommend that you add the more standard and accepted fatigue criterion, median frequency. 

The presentation of your results, both in figures and text requires additional work.  I have added some comments and suggestions below, in bold text.

Authors: Thank you. As we detailed in the manuscript “Classical EMG processing used to identify muscle fatigue using surface electrodes is based on the detection of a decrease in the median frequency (MF) of the EMG spectrum [48–50] during an isometric muscle action”. (lines 186-188).This paragraph has been reinforced by a new references:.

Allison, G.T.; Fujiwara, T. The relationship between EMG median frequency and low frequency band amplitude changes at different levels of muscle capacity. Clin Biomech (Bristol, Avon) 2002, 17, 464–469. De Luca, C.J.; Sabbahi, M.A.; Roy, S.H. Median frequency of the myoelectric signal. Effects of hand dominance. Eur J Appl Physiol Occup Physiol 1986, 55, 457–464.

Therefore, median frequency is considered as a reference measure for muscle fatigue in the literature. However, fatigue analysis can also be done by our approach based on spectral centroid, and this is reflected in previous literature. For better comprehension, we clarified this information with a new reference. The previous mentioned paragraph is followed by the next one:

“In fact, MF and spectral centroid (SC) are both considered measures of central tendency of the spectral distribution [51].

New Reference. 51. Chiang, S.; Vankov, E.R.; Yeh, H.J.; Guindani, M.; Vannucci, M.; Haneef, Z.; Stern, J.M. Temporal and spectral characteristics of dynamic functional connectivity between resting-state networks reveal information beyond static connectivity. PLoS One 2018, 13.

This explanation has been reinforced in the discussion section, as follows:

“As explained in the introduction, previous literature has shown that fatigue can be detected by observing a downward shift in the MNF or MFD. The present approach is based on a similar measure, namely spectral (SC), which measures the center of gravity of the power spectrum [51]. Moreover, the method implemented in the current study is based on computation of the first derivative of the SC corresponding to its variation slope. Hence, positive values of the slope indicate an upward shift of the SC whereas negative slopes indicate a downward shift of the SC. Consequently, the point at which the slope changes its sign (from positive to negative) indicates fatigue onset. Detecting muscle fatigue at second 12 through EMG, considered the gold standard [20–23], assured that the subject got fatigued during this task. Therefore, changes in acceleration after the second 12 guaranteed to occur due to fatigue“ (lines 271-280).

It is still not clear that the measure used to determine fatigue here, spectral centroid, has been previously validated.  Please add a reference to show this.

Your reference to second 12 is still jarring - surely there should be some error measure on this?

Regarding figures, in response to comment #3 from reviewer 1, we explain all the changes made, including a new figure 2 and modifying figure 3, which concur with some editor´s suggestion. Furthermore, a new video has been included as supplementary file.

Fig. 1 and Fig. 2 surely should be merged into one figure.

Fig. 2 also needs more detail in the caption – a quick definition of:  AR, ES, RF, VM, TA, SO…

Fig. 3 – you have increased the font size of the muscle names, but the font on the x and y axes are too small. You should re-do this figure, so that the message (there is a spectral shift at about 12 seconds in certain muscles, but not in others?) is clearer. In the caption, it is still not clear that this is averaged data over all trials.

Fig. 4 – could this not be merged with Fig. 3? i.e. add a new channel illustrating acceleration? 

Furthermore, we have included information in the last part from introduction section (related work suggested by reviewer 1), preprocessing (lines 174-176;179-180), data processing (188-189, new references 50 and 51), results (figure 3 explanation, and fatigue from acceleration in lines 251-256) and discussion section.

After all details given, authors hope results are easier to interpret.

There are still issues with your grammar/English, which are making the manuscript difficult to read. For example:

“Through rectified process, the average frequency is eliminated to start from value 0. In the filtered process, the low-pass 4th order Butterworth filter allows eliminating interferences or 178 components that are outside the bands of interest”

This should be written something like this:

“Through the rectification process, the average frequency is subtracted so that the mean value of the signal is zero (is this what you mean?). In the filtering process, the low-pass 4th order Butterworth filter removes interferences or components outside the bands of interest”.

Page 2, Line 57: Qualitative should be Quantitative here I think.

Authors: Thank you. The term “qualitative” refers to “how the movements were carried” (line 54) in others terms different from number of repetitions (quantitative). This can be better understand by reading previous lines “Lately, several studies have shown that besides quantifying motion, it is important to provide quality information about how the motion is carried out through kinematic parameters” (lines 45-47).

You are still trying to quantify (apply a numeric value to) how the movement were carried out. I would recommend rephrasing to “to provide additional information” on line 45.

Page 3, Line 94: "side was recorded at a frequency of 10.000Hz." - was should be were... and 10.000Hz reads as 10Hz to me, should this be 10 kHz? or 1000 Hz?

Authors: Thank you for this comment. It was a typing error. It has been replaced by 1.000Hz

The use of decimal point here is confusing for readers from English speaking countries – we use a comma.  If you change it to 1 kHz, or 1000 Hz, there will be no confusion.

Figure 3: I would recommend making muscle names clearer in the figure. In the caption, medial gastrocnemius (MG) is used, whereas GM is used in the Figure. "Represented along the time" does not make sense... 'Slope sign' is confusing also, the data looks all positive to me? So it is a value rather than a sign? Finally - is this data from one test? If so, this should be stated.

The data represent the average from 8 trials, as explained in methods section. This has been stated and clarified in that paragraph, as follows: “Each channel represent SC from averaged 8 measurements after being rectified, filtered and temporarily aligned” (lines 229-230)

Figure 4: This looks like representative data again, and should be stated in the caption.

Authors: Thank you. We have added in the caption the representative data relative to this figure and that was missing: “The cut-off time at second 12 corresponds to a descriptive acceleration value of 0.98 m/s2”

Figure 3 and Figure 4 seem repetitive.

Authors: Thank you. In this new version of the manuscript, we have added more information regarding figure 3, as follows:

As mentioned previously, these figures are still below the quality needed. Their individual importance to the message of your paper is not clear. 

“Each channel represent SC from averaged 8 measurements after being rectified, filtered and temporarily aligned”(lines 229-230).

Along with information added in caption from figure 4, we hope it gets easier to understand that figure 3 correspond to EMG SC slope in all 8 channels (8 muscles), from which channel 3 correspond to vastus medialis of the quadriceps, the one represented in figure 4 together with descriptive values of acceleration.

Line 204: 'The area under the curve for 204 the VM muscle signal decreases at second 12 with a slope of 12%.'

Authors: Thank you. Thank you. In the new version of the manuscript, that line is preceded by “In channel 3, which correspond to the VM muscle, a highlighted frequency shift between seconds 10 and 15 can be observed” (lines 232-233).

Line 226: 'The percentage of relative energy from acceleration showed an increase at second 19.' - surely there should be a mean and standard deviation on this time results? As it should be averaged across all trials? Some sort of accuracy measure on this result is needed.

Authors: Thank you so much for this question. Data from the percentage of relative energy from acceleration and its representation in figure 6 are calculated from temporal averaged signal (as can be read in preprocessing section: “Once EMG and acceleration signals were rectified and filtered, signals obtained during eight repetitions of the experiment were averaged in order to search for a statistically significant pattern. This requires the temporal alignment of signals acquired during the 8 different experiments, which was addressed by taking the maximum value of cross-correlation with respect to the first acquisition as a reference. This way, further processing was carried out using the averaged signals” ,lines 179-184). Therefore, mean and standard deviation could be calculated in the time-domain graph, but not in the frequency spectrum, as done in acceleration data.

“were averaged in order to search for a statistically significant pattern”:  This suggests that your analysis was biased – i.e. you were not open to not finding any pattern.  I would recommend rephrasing this to:   “were averaged, and this averaged signal was used for all further analysis”.

Author Response

Authors: As this second review made references to responses from the previous review, in this itemized point by point response, we have kept text from the previous review between squared brackets, so it is easier to differentiate from responses in this review.

Reviewer #2:

Many of my concerns from the previous review still stand. 

You have not adequately justified your criterion measure – I still recommend that you add the more standard and accepted fatigue criterion, median frequency. 

The presentation of your results, both in figures and text requires additional work.  I have added some comments and suggestions below, in bold text.

[Authors: Thank you. As we detailed in the manuscript “Classical EMG processing used to identify muscle fatigue using surface electrodes is based on the detection of a decrease in the median frequency (MF) of the EMG spectrum [48–50] during an isometric muscle action”. (lines 186-188).This paragraph has been reinforced by a new references:.

Allison, G.T.; Fujiwara, T. The relationship between EMG median frequency and low frequency band amplitude changes at different levels of muscle capacity. Clin Biomech (Bristol, Avon) 2002, 17, 464–469. De Luca, C.J.; Sabbahi, M.A.; Roy, S.H. Median frequency of the myoelectric signal. Effects of hand dominance. Eur J Appl Physiol Occup Physiol 1986, 55, 457–464.

Therefore, median frequency is considered as a reference measure for muscle fatigue in the literature. However, fatigue analysis can also be done by our approach based on spectral centroid, and this is reflected in previous literature. For better comprehension, we clarified this information with a new reference. The previous mentioned paragraph is followed by the next one:

“In fact, MF and spectral centroid (SC) are both considered measures of central tendency of the spectral distribution [51].

New Reference. 51. Chiang, S.; Vankov, E.R.; Yeh, H.J.; Guindani, M.; Vannucci, M.; Haneef, Z.; Stern, J.M. Temporal and spectral characteristics of dynamic functional connectivity between resting-state networks reveal information beyond static connectivity. PLoS One 2018, 13.

This explanation has been reinforced in the discussion section, as follows:

“As explained in the introduction, previous literature has shown that fatigue can be detected by observing a downward shift in the MNF or MFD. The present approach is based on a similar measure, namely spectral (SC), which measures the center of gravity of the power spectrum [51]. Moreover, the method implemented in the current study is based on computation of the first derivative of the SC corresponding to its variation slope. Hence, positive values of the slope indicate an upward shift of the SC whereas negative slopes indicate a downward shift of the SC. Consequently, the point at which the slope changes its sign (from positive to negative) indicates fatigue onset. Detecting muscle fatigue at second 12 through EMG, considered the gold standard [20–23], assured that the subject got fatigued during this task. Therefore, changes in acceleration after the second 12 guaranteed to occur due to fatigue“ (lines 271-280).]

It is still not clear that the measure used to determine fatigue here, spectral centroid, has been previously validated.  Please add a reference to show this.

Your reference to second 12 is still jarring - surely there should be some error measure on this?

Authors: Thank you. As explained in references 48-50, and new reference 51, fatigue is detected by a decrease in the median frequency of the EMG spectrum. However, spectral centroid can be also used to detect fatigue since a decrease in the median frequency implies a decrease in the spectrum centroid.

To show this, authors included a new figure 3, as well as median frequency values.

(see figure 3)

Figure 3 depicts the variation of both, the spectral centroid and median frequency of the EMG signal measured during the STS exercise.

Figure 3. EMG power spectrum obtained for the initial time window (at the beginning of the exercise) and the final time window (the end of the exercise). As it is shown, a decrease in both, the median frequency and the spectral centroid occurs during the exercise.

Regarding median frequency, its initial value was 51.5 Hz and the final value was 46.5 Hz.

This information has been provided in the manuscript, as follows:

“Fatigue is usually detected by decrease in the MDF of the EMG spectrum. In the case of VM muscle (channel 3), the MDF was 51.5Hz at the beginning and 46.5 Hz at the end of the 30-STS. SC can be also used to detect fatigue since a decrease in the MDF implies a decrease in the SC. To show this, Figure 3 depicts the variation of both, SC and MDF of the EMG signal of the VM muscle (channel 3) measured during 30-STS.

(figure 3 here)

Figure 3. EMG power spectrum obtained for the initial time window (at the beginning of the 30-STS) and the final time window (the end of the 30-STS). As it is shown, a decrease in both, the MDF and the SC occurs during the exercise.” 

(results section,page 7, lines 247-254)

This information was also contemplated in the discussion section

“As shown in figure 3, both measures can detect fatigue by a decrease in the MDF and SC.” (page 9, lines 289-290)

New reference:

Ma’as, M.D.F.; Masitoh; Azmi, A.Z.U.; Suprijanto Real-time muscle fatigue monitoring based on median frequency of electromyography signal. In Proceedings of the 2017 5th International Conference on Instrumentation, Control, and Automation (ICA); 2017; pp. 135–139.

[Regarding figures, in response to comment #3 from reviewer 1, we explain all the changes made, including a new figure 2 and modifying figure 3, which concur with some editor´s suggestion. Furthermore, a new video has been included as supplementary file.]

Fig. 1 and Fig. 2 surely should be merged into one figure.  

Authors: Thank you. As you propose, both figures have been merged into one singe figure 1.

Fig. 2 also needs more detail in the caption – a quick definition of:  AR, ES, RF, VM, TA, SO…

Authors: Thank you. Figure (now figure 1) caption now appears as follows: “Figure 1 X, Y and Z axes and their direction from a Smartphone placed in Sternum (left) and EMG sensor placement (right) in gastrocnemius medialis (GM), the biceps femoris (BF), the vastus medialis of the quadriceps (VM), the abdominal rectus (AR), the erector spinae (ES), the rectus femoris (RF), the soleus (SO) and the tibialis anterior (TA)..”(lines 137-140)

Fig. 3 – you have increased the font size of the muscle names, but the font on the x and y axes are too small. You should re-do this figure, so that the message (there is a spectral shift at about 12 seconds in certain muscles, but not in others?) is clearer. In the caption, it is still not clear that this is averaged data over all trials.

Authors: Thank you. As you proposed, we have re-done figure 3 (now figure 2), so the font on the x and y axes are bigger. Furthermore, in its caption we have added information to make clear that data represented are from EMG averaged data, “SC variation from EMG pre-processed and averaged signals from 8 measurements. Each channel represents signal from”… (lines 241-242). This is detailed in results section thank you to your comment in the previous review, but author did not include the information in the figure caption. We think that it is clearer now.

Regarding your question about spectral shifts, there are changes in slope around second 12 in other muscles but also around the whole duration of the test, as can be observed in the new figure 2. However, the most remarkable change in slope is the one that appears in channel 3 (vastus medialis of quadriceps) at second 12. To clarify this, we have rewritten results section, as follos:

Changes in slope were observed in all channels during the test. However, in channel 3, which correspond to the VM muscle, a highlighted frequency shift between seconds 10 and 15 can be observed.-..” (lines 236-238)

Fig. 4 – could this not be merged with Fig. 3? i.e. add a new channel illustrating acceleration? 

 Authors: Thank you. Authors think that figure 5 (which was figure 3) is easier to interpret. If acceleration was illustrated in figure 3 among all channels, it would appear at the bottom of the figure, instead of next to EMG from vastus medialis, like in figure 3. However, to make clear that EMG data from figure 3 represents the same data from channel 3 (vastus mediales), we have added between brackets this information: “Figure 4. EMG SC slope from VM muscle signal (channel 3) and trunk acceleration during 30-STS.” (lines 259-260).

Also, due to changes made for including new information from the response to your comment #1, this has also been clarified in the text prior figure 4, as follows:

“The highlighted frequency shift found in SC in VM muscle at second 12 (figure 2, channel 3), temporarily corresponds to a descriptive acceleration value of 0.98 m/s2. More details are shown in figure 4.” (lines 255-257)

[Furthermore, we have included information in the last part from introduction section (related work suggested by reviewer 1), preprocessing (lines 174-176;179-180), data processing (188-189, new references 50 and 51), results (figure 3 explanation, and fatigue from acceleration in lines 251-256) and discussion section.

After all details given, authors hope results are easier to interpret.]

There are still issues with your grammar/English, which are making the manuscript difficult to read. For example:

“Through rectified process, the average frequency is eliminated to start from value 0. In the filtered process, the low-pass 4th order Butterworth filter allows eliminating interferences or 178 components that are outside the bands of interest”

This should be written something like this:

“Through the rectification process, the average frequency is subtracted so that the mean value of the signal is zero (is this what you mean?). In the filtering process, the low-pass 4th order Butterworth filter removes interferences or components outside the bands of interest”.

  Authors: Thank you so much. That paragraph has been fixed as you proposed.

[Page 2, Line 57: Qualitative should be Quantitative here I think.

Authors: Thank you. The term “qualitative” refers to “how the movements were carried” (line 54) in others terms different from number of repetitions (quantitative). This can be better understand by reading previous lines “Lately, several studies have shown that besides quantifying motion, it is important to provide quality information about how the motion is carried out through kinematic parameters” (lines 45-47).]

You are still trying to quantify (apply a numeric value to) how the movement were carried out. I would recommend rephrasing to “to provide additional information” on line 45.

 Authors: Thank you so much. We have replaced the term “qualitative” by “additional”. We think that the proposed term fix better to express the idea in the introduction section.

[Page 3, Line 94: "side was recorded at a frequency of 10.000Hz." - was should be were... and 10.000Hz reads as 10Hz to me, should this be 10 kHz? or 1000 Hz?

Authors: Thank you for this comment. It was a typing error. It has been replaced by 1.000Hz]

The use of decimal point here is confusing for readers from English speaking countries – we use a comma.  If you change it to 1 kHz, or 1000 Hz, there will be no confusion.

  Authors: Thank you so much. We have eliminated the “dot” from 1.000, so now it can be read “1000 Hz”.

[Figure 3: I would recommend making muscle names clearer in the figure. In the caption, medial gastrocnemius (MG) is used, whereas GM is used in the Figure. "Represented along the time" does not make sense... 'Slope sign' is confusing also, the data looks all positive to me? So it is a value rather than a sign? Finally - is this data from one test? If so, this should be stated.

The data represent the average from 8 trials, as explained in methods section. This has been stated and clarified in that paragraph, as follows: “Each channel represent SC from averaged 8 measurements after being rectified, filtered and temporarily aligned” (lines 229-230)

Figure 4: This looks like representative data again, and should be stated in the caption.

 Authors: Thank you. We have added in the caption the representative data relative to this figure and that was missing: “The cut-off time at second 12 corresponds to a descriptive acceleration value of 0.98 m/s2”

Figure 3 and Figure 4 seem repetitive.

Authors: Thank you. In this new version of the manuscript, we have added more information regarding figure 3, as follows:]

As mentioned previously, these figures are still below the quality needed. Their individual importance to the message of your paper is not clear. 

 Authors: Thank you. As this comment is related to prior comments about figures 1-4, in previous response from the present review, it can be read changes made along figures and their caption to improve them.

[ “Each channel represent SC from averaged 8 measurements after being rectified, filtered and temporarily aligned”(lines 229-230).

Along with information added in caption from figure 4, we hope it gets easier to understand that figure 3 correspond to EMG SC slope in all 8 channels (8 muscles), from which channel 3 correspond to vastus medialis of the quadriceps, the one represented in figure 4 together with descriptive values of acceleration.

Line 204: 'The area under the curve for 204 the VM muscle signal decreases at second 12 with a slope of 12%.'

Authors: Thank you. Thank you. In the new version of the manuscript, that line is preceded by “In channel 3, which correspond to the VM muscle, a highlighted frequency shift between seconds 10 and 15 can be observed” (lines 232-233).

Line 226: 'The percentage of relative energy from acceleration showed an increase at second 19.' - surely there should be a mean and standard deviation on this time results? As it should be averaged across all trials? Some sort of accuracy measure on this result is needed.

 Authors: Thank you so much for this question. Data from the percentage of relative energy from acceleration and its representation in figure 6 are calculated from temporal averaged signal (as can be read in preprocessing section: “Once EMG and acceleration signals were rectified and filtered, signals obtained during eight repetitions of the experiment were averaged in order to search for a statistically significant pattern. This requires the temporal alignment of signals acquired during the 8 different experiments, which was addressed by taking the maximum value of cross-correlation with respect to the first acquisition as a reference. This way, further processing was carried out using the averaged signals” ,lines 179-184). Therefore, mean and standard deviation could be calculated in the time-domain graph, but not in the frequency spectrum, as done in acceleration data. ]

“were averaged in order to search for a statistically significant pattern”:  This suggests that your analysis was biased – i.e. you were not open to not finding any pattern.  I would recommend rephrasing this to:   “were averaged, and this averaged signal was used for all further analysis”.

Authors: Thank you so much for this comment as well as for previous comments done for improving the writing, so the manuscript is better expressed. We have rephrased that line as you proposed.

Reviewer 3 Report

This paper can be accepted and published in its current form。

Author Response

Reviewer #3:

This paper can be accepted and published in its current form

Authors: Thank you for your contributions made in the previous review.

Round 3

Reviewer 2 Report

Your paper is much improved, however you should still ask a native English speaker to proofread before publication.  Minor adjustments to the language would improve the impact of your paper, and the clarity of it's message. 

One last minor comment: 

As far as I can see, EDR is not defined in text, or in the caption of Figure 5.